# Ensemble Learning for Threat Classification in Network Intrusion Detection on a Security Monitoring System for Renewable Energy †

**Hsiao-Chung Lin** [1] , **Ping Wang** [1,*] , **Kuo-Ming Chao** [2] , **Wen-Hui Lin** [1] **and Zong-Yu Yang** [1]

[1] Green Energy Technology Research Center, Faculty of Department of Information Management, Kun Shan University, Tainan 710303, Taiwan; fordlin@mail.ksu.edu.tw (H.-C.L.); linwh@mail.ksu.edu.tw (W.-H.L.); s109000200@g.ksu.edu.tw (Z.-Y.Y.)

[2] Engineering and Computing, School of MIS, Coventry University, Coventry CV1 5FB, UK; csx240@coventry.ac.uk

\* Correspondence: pingwang@mail.ksu.edu.tw; Tel.: +886-6-2052139

† This paper is an extended version of our paper published in 3rd IEEE Eurasia Conference on IOT, Communication and Engineering 2021(IEEE ECICE2021), Yunlin, Taiwan, 29–31 October 2021.

**Abstract:** Most approaches for detecting network attacks involve threat analyses to match the attack to potential malicious profiles using behavioral analysis techniques in conjunction with packet collection, filtering, and feature comparison. Experts in information security are often required to study these threats, and judging new types of threats accurately in real time is often impossible. Detecting legitimate or malicious connections using protocol analysis is difficult; therefore, machine learning-based function modules can be added to intrusion detection systems to assist experts in accurately judging threat categories by analyzing the threat and learning its characteristics. In this paper, an ensemble learning scheme based on a revised random forest algorithm is proposed for a security monitoring system in the domain of renewable energy to categorize network threats in a network intrusion detection system. To reduce classification error for minority classes of experimental data in model training, the synthetic minority oversampling technique scheme (SMOTE) was formulated to re-balance the original data sets by altering the number of data points for minority class to imbue the experimental data set. The classification performance of the proposed classifier in threat classification when the data set is unbalanced was experimentally verified in terms of accuracy, precision, recall, and F1-score on the UNSW-NB15 and CSE-CIC-IDS 2018 data sets. A cross-validation scheme featuring support vector machines was used to compare classification accuracies.

**Keywords:** intrusion detection; ensemble learning; random forest algorithm; SMOTE; F1 score

## 1. Introduction

Cybersecurity mechanisms, such as network intrusion detection systems (NIDSs) and firewalls, detect network attacks and prevent hackers from gaining entry into the enterprise network. Most methods for intrusion detection focus on large-scale targeted cyberattacks, such as distributed denial of service (DoS) attacks [1], botnet attacks [2], ransomware attacks [3], phishing attacks [4], and credential theft [5]. Thus, studies on network attack detection have primarily focused on the use of specific security mechanisms as entry points into the enterprise network to defend against network threats. In practice, information security experts are often required to analyze and classify the threat type in cases of network intrusion, and it is often impossible to judge new types or variants of cyber threats in real time. For the implementation of network security applications, NIDSs must support intrusion detection when the volume of flow inspections is large by performing anomaly detection and exploring new threats through machine learning (ML) algorithms.

Typically, network intrusion detection (NID) involves collecting behavioral information to classify all potential threats into attacker and victim categories under some given

constraints on the quantity of packets collected and NIDS computational time. Many classification approaches incorporate ML algorithms to assist managers in precisely identifying network attacks [4–7]. ML techniques for threat classification—such as support vector machine (SVM) and hybrid approaches—are used to aid category prediction, wherein the SVM [4] is incorporated with other classification approaches, such as those based on decision trees (DTs) [5], principal component analysis (PCA) [6], and the Dempster–Shafer theory [7]. These network threat classification schemes are summarized in Table 1.

**Table 1.** Machine learning approaches for network threat classification.

|  | **Features** | **Contributions and Experimental Results** |
|---|---|---|
| SVM Guan et al. [4] | Four support vector machine (SVM) classifiers are used to categorize network data into five classes: denial of service, probe, U2R, R2L, and normal. | The agent and SVM were used to improve the detection precision of intrusive attacks for network intrusion detection (NID). |
| Fuzzy multiclass SVM Li et al. [5] | A decision tree (DT) is constructed using fuzzy multiclass SVM in which each data class is assigned a fuzzy membership during training to reduce the effects of outliers and response time. | The combined fuzzy theory and multiclass SVM improved detection accuracy and reduced training time. |
| SVM with PCA Kausar et al. [6] | The method is used for feature transformation into higher dimensions for determining the feature subset, after which the performance (in terms of detection rate and rate of false alarms) can be determined during testing. | The use of reduced features in training the support vector classifier accelerated the learning of normal and intrusive patterns. Improved accuracy (99.465%) and false alarm rate (0.525%) were observed for a subset of 10 features. |
| SVM with Belief theory Singh et al. [7] | This method is a hybrid one wherein intrusive behavior is detected using the Dempster belief algorithm (DCA) and Dendritic Cell Algorithm and where data are classified with the SVM. | The detection rate from the joint use of DCA and SVM was less than 92% (by contrast, the method proposed in the present paper reached 96%). |

Because information diversification services produce diverse and complex threat patterns, a single classifier in an ML model may not produce perfect predictions for a given data set under certain real-time requirements for intrusion detection by an NIDS. As attacks on large-scale networks become more diverse, a basic classifier in ML models, such as SVM, becomes increasingly unable to effectively process a large volume of traffic in large-scale networks with complex intrusion patterns. Therefore, ensemble learning-based techniques, such as random forest (RF) [8–10], boosting [11], gradient boost DT (GBDT) [12], and stacking [13], are adopted to help security managers detect complex threats from a variety of sources.

In practice, classes necessarily have an imbalanced distribution in information flows because the volume of traffic is large and because certain types of anomalies occur at a low frequency. Thus, the classification performance of supervised ML techniques, such as DT, naive Bayes, and SVM techniques, are affected by imbalances in the number of data points for each threat class in a given data set.

Intrusion detection schemes have tended to ignore the magnitude of imbalanced data and overfitting-related difficulties in threat data. In imbalanced intrusion data sets, ML algorithms intuitively provide more accurate predictions for classes with many data points (i.e., majority classes) and less accurate predictions for classes with a small number of data points [14]. To maintain prediction stability at high precision, rebalancing sampling must be applied to the raw data to achieve a balance in the number of data points for each class.

Furthermore, threat data sets necessarily include noise extracted from flow traffic; such noise causes overfitting in model training. An overfitted model predicts future observations

poorly because it contains more parameters than the data can justify. Practically, ensemble learning algorithms, such as RF, can overcome overfitting in threat data in which RF corrects for overfitting from the DTs.

Inspired by Ho's study [9], the present study proposes an RF-based ensemble learning algorithm associated with a uniform distribution resampling scheme for minority classes based on a Synthetic Minority Oversampling TEchnique (SMOTE) for NIDs. SMOTE manages the imbalance between threat classes in the data set in advance. Moreover, eliminate the irrelevant and unwanted features from the dataset in intrusion detection causes to possibly faster and more accurate detection. Consequently, feature selection scheme C4.5 is incorporated with RF to reduce the number of input variables to mitigate overfitting in developed predictive models. The performance of the proposed algorithm was demonstrated through experiments on the UNSW-NB15 data set [15] and CSE-CIC-IDS 2018 [16] for threat classification. A cross-validation scheme with a support vector classifier (SVC) was used for performance comparisons.

In summary, the primary contributions of this study are as follows:

- Multiclass threat classification was achieved using the RF method. Moreover, sources of open intrusion attacks in the UNSW-NB15 and CSE-CIC-IDS 2018 data sets were accurately classified, indicating our method's ability to classify threats as part of an NIDS;
- To improve the performance of random forests, the RF is incorporated with C4.5 algorithm to dimension reduction of training data that accelerates the training time of high-dimensional data in the model training;
- To improve data imbalanced situation, the resampling process of the SMOTE algorithm is proposed to reduce the skew in the distributions of classes by modifying the number of instances for minority class;
- The accuracy of the proposed algorithm was 99.81% for two-class classification of UNSW-NB15 and 87.64% for multiclass classification;
- The classification accuracy of intrusion detection was 99.98%% for two subcategories of CSE-CIC-IDS 2018 and 96.53% for six subcategories of classification accuracy;
- Compared with the classification accuracy of competing approaches on UNSW-NB15, such as [14,17], the proposed RF-NID algorithm performed better in threat class identification in cases where threats stemmed from multiple sources.

The remainder of this paper is organized as follows: Section 2 presents a literature review, Section 3 presents an analytical model of NID, Section 4 details the evaluation of this paper's method, and Section 5 concludes the paper.

## 2. Overview of SMOTE Schemes and Ensemble Learning Schemes

This section reviews methods for addressing the problem of imbalanced threat data in training and introduces a method where an RF is used for ensemble learning to classify possible attacks.

### 2.1. SMOTE Techniques for Imbalanced Data

An imbalanced data set is one where the minority class is greatly outnumbered by the majority class with respect to their number of data points (Figure 1). This skew in the distributions of classes makes classification for the minority class imprecise. This problem is serious because predictions for the minority class are typically the most crucial [18]. Additionally, such skewness makes training less effective.

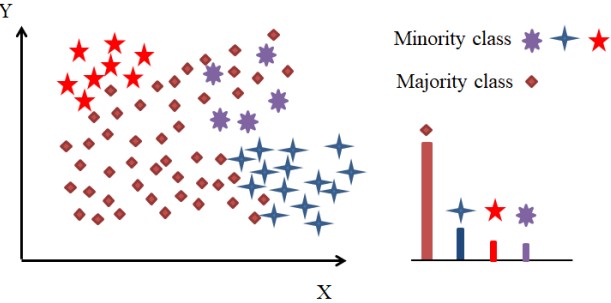

**Figure 1.** A case of unbalanced data.

One approach to addressing this class imbalance is to randomly resample the raw data for training. This approach can be divided into three subapproaches that all involve deleting examples from the majority class. Undersampling, the first subapproach, is where extra examples are screened from the majority class. Oversampling, the second subapproach, is used to duplicate examples from the minority class. For example, the SMOTE scheme [18] balances the minority class of raw threat data, adjusts them to the other categories, and improves the imbalance in the minority classes. Uniform sampling, the third approach, is used to modify the structure of the data for each threat class.

For example, the number of data points for each threat category in the UNSW-NB15 data set is detailed in Table 2. The data set has a considerable class imbalance, as illustrated in Table 2. For example, the 'generic' class accounts for 22.81% of the total data, whereas the categories with the smallest number of data points, 'shellcode' and 'worms', only account for 0.65% and 0.007% of the total data, respectively. Minority class-like analysis (1.14%), backdoor (1.00%), shellcode (0.65%), and worms (0.07%) have a smaller number of data points, as presented in Table 2. Hence, oversampling is used to increase the number of data points for these classes.

**Table 2.** Number of data points for each threat category (UNSW-NB15).

| Threat Category | Record No. of Training Data | Record No. of Test Data |
|---|---|---|
| Normal | 56,000 (31.94%) | 7000 (44.94%) |
| Generic | 40,000 (22.81%) | 8871 (22.92%) |
| Exploits | 33,393 (19.04%) | 1132 (13.52%) |
| Fuzzers | 18,184 (10.37%) | 6062 (7.36%) |
| DoS | 12,264 (6.99%) | 4089 (4.97%) |
| Reconnaissance | 10,491 (5.98%) | 3496 (4.25%) |
| Analysis | 2000 (1.14%) | 667 (0.81%) |
| Backdoors | 1746 (1.00%) | 583 (0.71%) |
| Shellcode | 1133 (0.65%) | 387 (0.47%) |
| Worms | 130 (0.07%) | 44 (0.05%) |

In imbalanced data (Table 2), the majority class, a data set class comprising more than half (50%) of the data set's data points, represents the main part of the instances labelled as one class, and the minority classes represent considerably fewer instances labelled $c_i$, where $i = 1, \ldots, r$. The symbol $c_i$ denotes the essential class of samples to be classified, and this class includes behavioral features and class labels. In [19], the imbalance ratio (IR) was defined as follows:

$$\text{Imbalance Ratio}(c) = \frac{|\text{Instances of Major Class}|}{|\text{Instances of Mainority Class}|}. \tag{1}$$

If the IRs of the sample data are high, the model's classification accuracy and prediction reliability are unstable and low [20]. The data points for minority class—represented as a pair (feature set, class label)—should be made to have a strong signal among each threat class.

SMOTE generates synthetic examples for minority classes to achieve enhanced performance in imbalanced data sets. As presented in Figure 2, the minority class was oversampled by taking each minority class sample and introducing synthetic examples along the line segments that join any of the *k* minority class' nearest neighbors. In other words, increased data for minority class are generated by two neighbors from the five nearest neighbors random and one sample is generated in the direction of each [18].

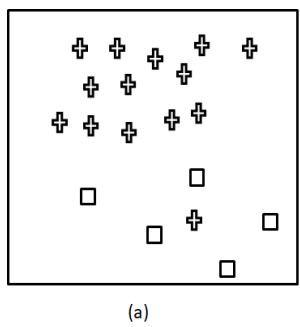 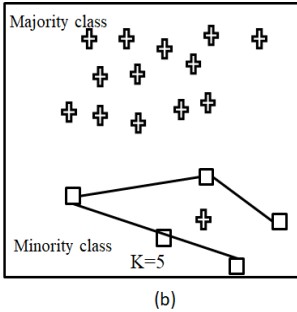 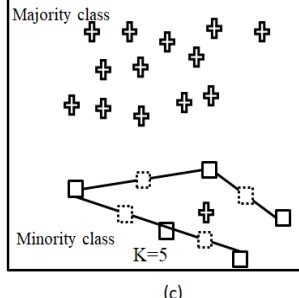

(a) (b) (c)

**Figure 2.** An example of resampling process of SMOTE. (**a**) Minority instances of the imbalanced data (**b**) Select five nearest neighbors (**c**) Generated two neighbors along the line segments.

Given a training data set $(X, Y)$ in which $X$ represents a data sample in minority class samples, $X = x_1, \dots, x_n$, $Y$ is the over-sampling target from $X$, and $Y = y_1, \dots, y_N$, $y_i$ represents the $i^{th}$ of the $N$ nearest neighbors of the $X$. Notably $N$ synthetic samples (i.e., $y_1$, $\dots, y_N$) are randomly selected from $K$ nearest neighbors as shown in Figure 2. To increase the number of minority class samples, the random interpolation operation between $X$ and $y_i$ ($i = 1, 2, \dots, N$) is performed by the following formula to obtain the interpolated sample $p_i$, [21]

$$p_i = X + rand(0,1) \times (y_i - X), \tag{2}$$

where $rand(0, 1)$ represents a random number in the range (0, 1). Obviously, the number of sampling augmentation $N$ depends on the IR of the dataset used. In further, $N = round(IR)$, where round(IR) represents the value obtained by rounding up the IR. Once achieved the above over-sampling operation, new synthetic samples are generated randomly from minority class samples and their neighbors, and the majority class samples and the minority class samples became balanced. Researchers have focused on using the SMOTE method as a data preprocessing mechanism for balancing data sets in intrusion detection (Table 3).

### 2.2. RF Algorithms

The RF scheme was created in 1995 [8] in which DTs are aggregated to improve the performance of a single DT. As shown in Figure 3, RFs comprise DTs on randomly selected training data sets; in this approach, predictions are obtained from each tree and the best solution is selected by means of majority votes in ensemble algorithms. RFs typically provide a fairly accurate indicator of the feature's importance [9].

The RF algorithm creates *n* DTs on randomly selected data points and synthesizes the prediction results from each DT (basic classifier) through group voting to reduce the variance of classification results. Put simply, an RF is an ensemble learning method that is better than a single DT because it reduces the overfitting outcomes by averaging the results of DTs (Figure 4).

**Table 3.** SMOTE with machine learning approaches for network threat classification.

| Study | Features | Contributions and Experimental Results |
|---|---|---|
| Chawla, Bowyer, Hall, Kegelmeyer (2002) [18] | • Synthetic minority oversampling technique (SMOTE), an oversampling method for balancing imbalanced data sets, is used to improve the accuracy of classifiers for a minority class. | • The SMOTE classifier outperformed an under-classifier, loss ratio-based classifier, and naive Bayes classifier. |
| Blagus and Lusa (2013) [22] | • This study investigated SMOTE from theoretical and empirical perspectives using simulated and empirical high-dimensional data. | • SMOTE improved the performance of k-NN classifiers for high-dimensional data when the number of variables was reduced using a variable selection method. |
| Zong, Chow, Susilo (2018) [14] | • A two-stage classifier approach for network intrusion detection systems (NIDSs) is applied to imbalanced data sets on intrusion detection; in this approach, minority and majority intrusion classes are separated in training and detection. | • Minority and majority intrusion classes are separated in training and detection to improve the overall detection rate of minority classes and to reduce the error rate for UNSW-NB15 data set. |
| Das, Khan, Saha (2019) [23] | • A rough random forest (RF) algorithm in conjunction with binarization techniques is used to decompose an original data set into subsets of binary classes to balance multiclass imbalanced data sets. | • The proposed method outperformed other methods featuring the Tree Bag Model and SMOTE+Tree Bag Model with respect to the receiver operating characteristic curve and corresponding area under the curve. |
| Tan et al. (2019) [21] | • Proposes a method of using the SMOTE to balance the dataset and then uses the random forest algorithm to train the classifier for intrusion detection. | • The simulations are conducted on a benchmark intrusion dataset KDDCup99, and the accuracy of the random forest algorithm has reached 92.39%, which is higher than other comparison algorithms. |
| Karatas, Demir, Sahingoz (2020) [19] | • This stud proposed six machine-learning-based intrusion detection systems with SMOTE for data classification in conjunction with k-nearest neighbor (k-NN), RF, gradient boosting, adaboost, DT, and linear discriminant analysis algorithms. | • In experiments, the proposed approach considerably increased the detection rate for rarely encountered intrusions in the CSE-CIC-IDS 2018 data set. |
| Hui, He, Ye, Zhang (2020) [20] | • Conduct the comparative experiments on analysis for the intrusion detection problems using Xgboost, Random Forest, Bagging, and Adaboost. | • Experimental results demonstrate that PSO-Xgboost model outperforms other comparative models in precision, recall, macro-average, and mean average precision (mAP) on NSL-KDD dataset. |
| Jun, Sheng, Wang (2020) [24] | • Combine the spatial feature and temporal feature, we fuse GBDT model and Gated Recurrent Unit (GRU) model to make a quadratic ensemble model as intrusion detection system. | • The experimental results show that the advanced spatial-temporal intrusion detection system based on ensemble learning achieves better accuracy, recall, precision and F1 score than the state-of-the-art methods on CIC-IDS-2017 dataset. |

<div align="center">Table 3. <em>Cont.</em></div>

| Study | Features | Contributions and Experimental Results |
|---|---|---|
| Kasongo, Sun. (2020) [17] | • Present a filter-based feature reduction technique using the XGBoost algorithm in conjunction with SVM, k-NN, Logistic Regression (LR), Artificial Neural Network (ANN) and DT on the UNSW- NB15 intrusion detection dataset. | • In the case of the DT classifier, the test accuracy has increased from 66.03% to 67.57% using the 42 and 19 features, respectively.<br>• For the multiclass classification scheme. Moreover, for the binary classification process, the DT has increased the test accuracy from 88.13% to 90.85% using the reduced feature dimensions of the UNSWNB15 respectively. |
| Wu et al. (2021) [25] | • Combining the K-means clustering with the SMOTE sampling algorithm to increase the number of minor samples and thus achieved a balanced data set. | • The performance was tested using the NSL-KDD dataset with a classification accuracy of 99.72% on the training set and 78.47% on the test set. |
| Luyao, Lu (2021) [26] | • Propose an intrusion detection model based on SMOTE and convolutional neural network (CNN) ensemble to solve the problem of imbalanced datasets. | • Evaluate the performance of the proposed model on NSL-KDD dataset to model decision-making show that the model's F1 score are better than traditional algorithms in the classes with few samples and improves the efficiency of network intrusion detection. |

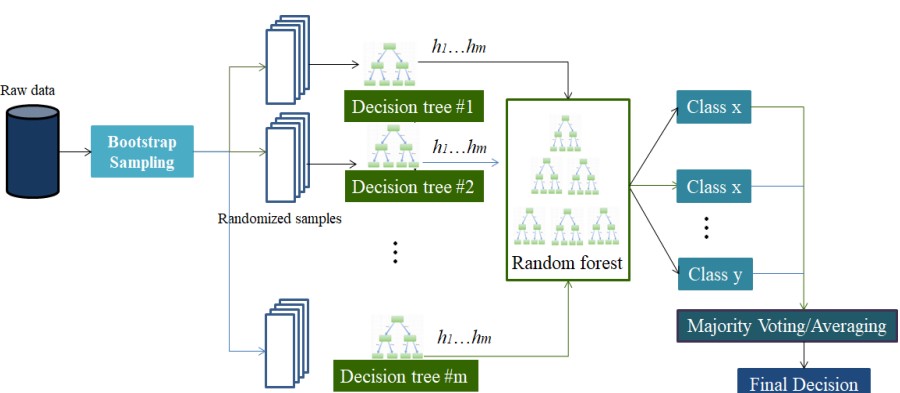

**Figure 3.** RF algorithm.

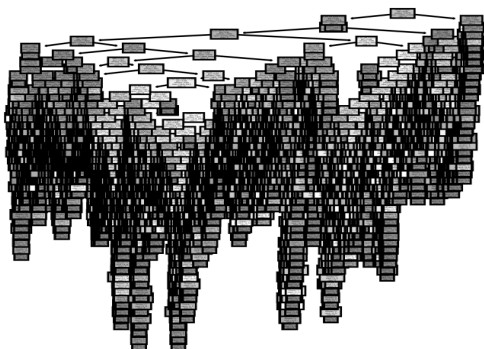

**Figure 4.** RF for the number of trees *n* = 100 in our study.

### 3. Application of Proposed RF Algorithm for Intrusion Detection

The proposed NID model combines the ensemble learning RF algorithm and the SMOTE to maintain high precision in prediction stability. The overall structure of the model is presented in Figure 5. The system classifies network traffic in three steps. In Step 1, data preprocessing is performed on the threat data set with the SMOTE scheme; this is done to augment the data of the minority class in the training set to balance the data. The balanced data points are then used for model training, during which the model classifies the potential attacks according to whether they are of normal traffic or malicious behavior. The raw data are analyzed and features are selected using the C4.5 algorithm; the algorithm reduces the number of input variables to both reduce the computational cost of the model and accelerate the classification performance of the model. In Step 2, RF ensemble learning is used to train component classifiers and aggregate the results of the component classifiers by randomly selecting subsets of the training data. A metaclassifier is trained with a majority voting approach to perform a threat classification of the observations. In Step 3, the NID model is evaluated in a large-scale network environment using two open-source data sets, UNSW-NB15 [15] and CSE-CIC-IDS2018 [16], on various threats.

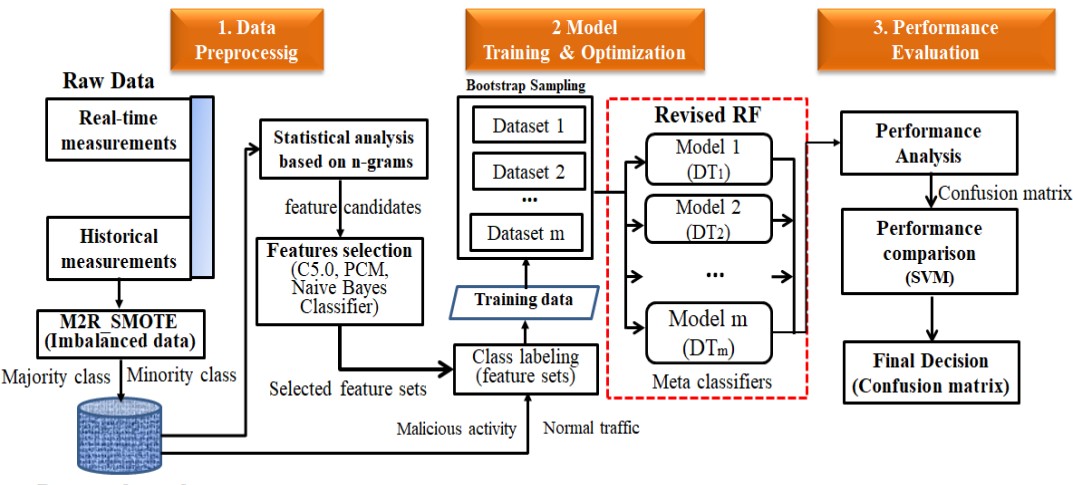

**Figure 5.** Diagram of developed NID model.

As illustrated in Figure 5, the threat analysis process comprises three subprocesses: data preprocessing (including resampling process and feature selection) model training and optimization, and performance evaluation for intrusion detection of network attacks as follows.

Step 1. Data preprocessing

Data preprocessing proceeds in the following steps: (1) encoding symbolic behavior (features) to numerical values, (2) normalising the scale, (3) resampling strategies for imbalanced data sets, and (4) feature selection.

Step 1.1. Normalization

Because the raw data are of various resolutions and ranges, the numerical data of each feature are normalized to a range of $[-1, 1]$ using the following min–max formula.

$$d = \frac{d - min_d}{max_d - min_d}. \tag{3}$$

Step 1.2. Resampling strategy

To improve the situation of class imbalance, the user must modify the number of instances for minority class. In other words, if parts of class of the minority class were misclassified in the training process, it indicates the number of instances for this class needs to be augmented to increase the classification accuracy of model training. By applying

Formula (4), new synthetic samples ($p_i$) and the resampling magnification of the dataset ($N$) are generated randomly from minority class samples and their neighbors using oversampling operation. In the next section, we detail our evaluation of this RF in intrusion detection classification.

Step 1.3. Feature selection

Feature selection is the process of reducing the number of input variables to develop a predictive model to improve the performance of the model by reducing the computational costs of modelling. The process of obtaining a reduced set of features to create a predictive model requires experimentation and substantive knowledge about the problem at hand. To create a model, the subset of selected features in our data set was determined in accordance with C4.5 DT theory in which the values of attributes are represented by branches and attributes are arranged as nodes according to the data classification in DT learning. The C4.5 algorithm begins with the original set $S$ as the root node. In each iteration of the solution procedure, the algorithm repeats itself through every unused attribute of the set $S$ and process the information to calculate the $IG(S)$ of that attribute.

The attribute with the largest information gain is used to split $S$ in the present iteration. Finally, the set $S$ is split by the selected attribute to produce data subsets. Let the information gain of attribute $A$ be represented as $IG(S,A)$. The measure of the difference in entropy before and after $S$ is split by $A$ is

$$IG(S, A) = E(S) - \sum_{i=1}^{n} p(S_i)E(S_i) = E(S) - \sum_{i=1}^{n} \frac{|S_i|}{|S|} E(S_i). \tag{4}$$

$$E(S) = - \sum_{i=1}^{n} p(S_i) \log_2 p(S_i), \tag{5}$$

where the entropy $E(S)$ is a measure of the amount of uncertainty in the data set $S$ [27,28], $S_i$ represents the subsets created from splitting set $S$ by attribute $A$ such that $k = 5$. $S = U_{i,} (s_i)$, $p(s_i)$ is the ratio of the number of elements in $S_i$ ($|S_i|$) to the number of elements in set $S$ ($|S|$), and $E(s_i)$ is the entropy of subset $s_i$. Equation (3) describes the uncertainty in $S$, which is reduced after set $S$ is split in terms of attribute $A$. Theoretically, the attribute that maximizes the difference is selected. In practice, however, choosing a suitable threshold value for determining the exact the number of features ($N_f$) extracted from the possible feature candidates is difficult.

However, information gain tends to be biased in favour of attributes with many distinct values, such as social ID, in the data set. Therefore, the information gain ratio (IGR) is selected as a measure to reduce bias towards multivalued attributes by accounting for the number and size of the branches when choosing an attribute. IGR corrects the information gain by considering the intrinsic information of a split ($Split\ Info_A(S)$) through the normalization process of the information gain [28] as follows.

$$Split\ Info_A(S) = - \sum_{i=1}^{n} \frac{|S_i|}{|S|} \log_2 \frac{|S_i|}{|S|}. \tag{6}$$

Features with a large amount of intrinsic information are less useful for classifying the data. In the C4.5 DT, the ratio of information gain of the attribute $IGR(A)$ is used to calculate the $IGR$ for each attribute to determine the relevant subset of features used in splitting in the tree based on the maximum gain in information for C4.5 DT. $IGR(A)$ is defined as

$$IGR(A) = \frac{IG(S, A)}{Split\ Info_A(S)}. \tag{7}$$

The C4.5 DT uses the $IGR$ measure, which is the information gain divided by the split information (S).

Step 2. Model training and optimization

In this phase, the RF-based classifier is trained to detect specific network attacks from the behavioral patterns, in the data, that are associated with families of real threats. RF classifiers obtain predictions from each DT and select the best solution by means of PV rule.

Step 2.1. Bootstrapping

To create DTs with randomly selected data samples, the bootstrapping method is applied to segment the training data set. This method is a resampling technique used to estimate statistics on a population by sampling a data set with replacement. In statistics, bootstrapping involves drawing sample data repeatedly with replacement from a data source to achieve an unbiased estimate of a population parameter [29].

Step 2.2. Model training using RF

Typically, four steps are included in the RF process.

Step RF.1—The bagging algorithm is used to randomly generate $n$ training data sets that are subsets of a given data set. A bootstrap sample is obtained from the original data through sampling with replacement. In the case of $n$ training samples, each sample has $M$ features and training data sets are randomly selected (but replaced) to form $n$ training data sets that are subsets of the data set.

Given a training data set $(X,Y)$ in which $X = x_1, ..., x_n$ and the target $Y = y_1, ..., y_n$, the bagging method executes bootstrap sampling $B$ times to construct multiple DT models, and the method then trains the DT model on the new samples. For example, for b = 1, ... , B, the sample is replaced with $n$ training examples from $X, Y$, noted as $X_b, Y_b$. A classification tree $f_b$ is then trained on $X_b$ and $Y_b$.

Step RF.2—A DT is created for every subset of the data set.

For each training set, a tree predictor $\theta_i$ is generated by randomly selecting $m$ features used for tree splitting from all features $M$ ($m < M$). Generally, the splitter with the smallest Gini index that generates $n$ classification and regression trees for classification purposes was selected through the RF approach. The Gini index is a number describing the quality of the split of a node on a feature. If data set $D$, for example, contains samples from $C$ classes, the Gini index is defined according to [10] as follows:

$$\text{Gini}\,(D) = 1 - \sum_{c=1}^{C} P_c{}^2, \tag{8}$$

where $P_c$ is the relative frequency of class $c$ in $D$.

Step RF.3—The prediction results from each DT are synthesized through a plurality vote (PV) that generally increases the classification accuracy of the overall model. The vote determines the class $i$ that maximizes the sum based on the majority voting rule.

$$\text{class}(x) = \text{Arg max}[\sum_{k} g(y_k(x), c_i], \tag{9}$$

where $x$ is the behavior feature of the sampling data, $y_k(x)$ is the classification result of the $k$th DT, and $g(y_k(x), c_i)$ is a counting function defined as

$$g(y,c) = \begin{cases} 1, & y = c \\ 0, & y \neq c \end{cases}, \tag{10}$$

where $g(y_k(x), c_i)$ is the prediction result of the classifier $i$ that $x$ belongs to class $k$. $y_k(x)=1$ for the true class $k$ of $x$; otherwise $y_k(x) = 0$.

Step RF.4—The outcome chosen by the most decision trees is the final indicator.

Subsequently, the behavioral patterns of network threats are classified into those associated with normal versus abnormal connections. Thereafter, the accuracy of RF classifiers in detecting existing or identified network attacks is evaluated using Equations (8)–(10).

Step 3. Performance evaluation

This step is performed to validate the classification performance of the proposed classifier on unbalanced data. In this study, we conducted such an evaluation in terms

of accuracy, precision, recall, and F1 score on the experimental data set. In particular, SMOTE with uniform distribution resampling was used in the training experiment. Finally, a cross-validation scheme was adopted to compare the predicted accuracy of the developed model with an SVC.

## 4. Results

The applicability of the proposed RF to behavioral classification in cases of imbalanced data were demonstrated through two examples of NID associated with a complete feature set($N_f$ = 42) and a reduced feature set($N_f$ = 23). The experiments were conducted in Python using the ML library of the scikit-learn package; this library is an open source library for classification algorithms, such as RF, SVM, and naive Bayes classifier, logistic regression, and quadratic discriminant analysis algorithms. The software used is described in Table 4. The software was run on an Intel Core i3-4160 dual core CPU clocked at 3.0 Ghz and 8 GB of DDR3 RAM; the operating system was Ubuntu Desktop 20.04.3 LTS, and the database platform was MongoDB 5.0.3. The experimental environment is depicted in Table 4.

**Table 4.** Experimental environment for RF-based security monitoring.

| Numerical and Machine Learning Library | |
| :---: | :---: |
| Python 3.8.10 | scikit-learn |
| | imbalanced-learn |
| | numpy |
| | scipy |
| | pandas |

### 4.1. Case I: Binary Classification and Multiclass Classification (UNSW-NB15)

This first case pertained to profiles of cyberattacks on Internet of Things devices on a cloud server. In such a case, security managers must constantly monitor and compare the statistical details of each flow entry between consecutive time windows. NID was executed in the following three phases: (1) data preprocessing, (2) model training and optimization, and (3) performance evaluation. The workflow of the security analysis is illustrated in Figure 5.

Step 1: Data Preprocessing Phase

In the intrusion detection experiment, the UNSW-NB15 data set [15,30] was selected as a comprehensive data set for examining the performance of the developed classifier. This data set was divided into training and testing sets. We selected the UNSW-NB15 data set because it had three advantages over similar data sets. First, it contains up-to-date behavioral features with contemporary attack sequences. Second, it involves a set of features from the payload and header of packets to reflect the network packets efficiently. Third, it contains many complicated features that the model can learn from to discriminate more accurately.

The UNSW-NB15 data set was created by the IXIA Perfect Storm tool in the Cyber Range Lab of the Australian Centre for Cyber Security to produce a hybrid data set of synthetic contemporary attack behaviors in real-world network traffic. The tcpdump tool was used to capture 100 GB of raw traffic (in Pcap files). This data set had nine attack categories, namely, fuzzers, analysis, backdoors, DoS, exploits, generic, reconnaissance, shellcode, and worms, and 42 features with a class label from 2,540,044 observations.

The training set contained 175,341 records (68.05%), and the testing set contained 82,332 records (31.95%) from malicious and normal files. The training set had 119,341 (68.06%) and 56,000 (31.94%) intrusion attack files and normal files, respectively. The testing set had 45,332 (55.06%) and 37,000 (44.94%) intrusion attack files and normal files, respectively. As indicated in Table 2, the UNSW-NB15 data set was imbalanced: it had a large difference in the number of data points between threat categories.

Step 1.1. Normalization

Of the 42 features in the UNSW-NB15 data set, 39 were numerical features and 3 were symbolic (attack class) features. We performed a symbol conversion of the network packets. First, the category attributes of proto, service, and state were converted to a numerical format through one-hot encoding with the function *get_dummies()* of the pandas software library.

Step 1.2. Resampling strategies

According to formula (2), new synthetic samples ($p_i$) and the resampling magnification of the dataset (*N*) are generated randomly from minority class samples and their neighbors using over-sampling operation. Typically, the number of each class in the data set must be balanced (i.e., the data set must have as close to a uniform distribution as possible) for a more precise model to be obtained. In [18], it is recommended that the number of data points for each minority class should be augmented to be 200% of the original number. Subsequently, through an application of Equation (3), raw data were scaled to the range of [−1, 1] based on the *minmax()* function.

Step 1.3. Feature selection

By applying Equations (4)–(7) to the UNSW-NB15 data set, we first ranked the features according to the scores assigned by the IGR measure. The set of reduced features selected from the top 23 of the 42 total features using the IGR approach are displayed in Table 5.

**Table 5.** Top 23 features by weight.

| Feature | Weighting | Rank | Feature | Weighting | Rank |
|---------|-----------|------|---------|-----------|------|
| Sttl | 0.1543 | 1 | Dmean | 0.0271 | 13 |
| ct_state_ttl | 0.0694 | 2 | Sinpkt | 0.0267 | 14 |
| Dload | 0.0578 | 3 | dbytes | 0.0250 | 15 |
| Dttl | 0.0527 | 4 | ct_dst_src_ltm | 0.0248 | 16 |
| Tcprtt | 0.0412 | 5 | smean | 0.0246 | 17 |
| Dur | 0.0378 | 6 | state_INT' | 0.0219 | 18 |
| Sload | 0.0366 | 7 | ct_srv_src | 0.0212 | 19 |
| Ackdat | 0.0351 | 8 | spkts | 0.0165 | 20 |
| Rate | 0.0306 | 9 | djit | 0.0145 | 21 |
| ct_srv_dst | 0.0305 | 10 | dloss | 0.0133 | 22 |
| Synack | 0.0303 | 11 | ct_dst_sport_ltm | 0.0125 | 23 |

Step 2. Model Training and Optimization Phase

In this step, the experiments were divided into two parts: (1) 42 behavioral patterns of the test samples were identified to conduct the training experiment and (2) ranked features using IGR rate (Table 5) were extracted to derive 23 features from 42 features to examine the RF-based model classification accuracy.

Step 2.1. Data bootstrapping

Bootstrapping was used to divide the training data set into *m* subsets to help the data analyzer create DTs with randomly selected data points.

Step 2.2. Model training using RF

In this study, the model was trained on 42 behavioral patterns before extracting 23 features that had the highest IGR rate (Table 5). The performance of the RF-based classification model was then evaluated using Equations (8)–(10) by training component classifiers and aggregate the results of component classifiers from randomly selecting sub-datasets of the training data, and finally trains a meta-classifier with majority voting approach to perform the threat classification of the samples. Using GridSearchCV to set for the model parameters of binary classification on Random Forest. The search parameters of RF are set as Table 6.

**Table 6.** Optimal parameter search for random forests using GridSearchCV.

| Parametert Model | $n$-Estimators | Max-Features | Max-Depth | Criterion Tree Split |
|---|---|---|---|---|
| RF | [50,100,150,200,500,1000] | Auto, sqrt | [4,5,6,7,8] | Gini |

Step 3. Performance Evaluation Phase

To obtain the optimal classification accuracy for RF model, it first needs to determine how many trees (component classifiers) to be generated in RF algorithm. Typical value for the number of trees in most cases is 10, 30, or 100. It is very few practical cases more than 300 trees that may increase the cost in computation time for learning these additional trees [31]. In this study, the experiments for deciding the appropriate number of tree on RF were conducted by the different numbers by examining the RF-based model classification accuracy: 25~200 trees ($n$ = 25~200) of the training samples. The low classification error of the RF algorithm, as presented in Figure 6, was based on data from approximately 100 to 200 trees.

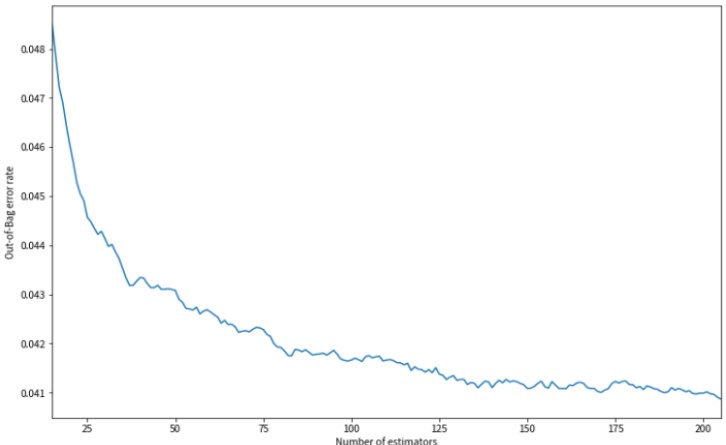

**Figure 6.** Binary classification error with a given number of trees.

From Table 7, it is seen that the there is a better performance of multi-classification (i.e., accuracy precision, recall, F1 Score, and ROC AUC) when $n$ =100, 150, 200, compared to $n$ = 50. Normally, more trees are generally better precision for RF algorithm; however, also increase the computational time of model. Notably, training with the training set for 200 trees, it gets about a 0.04% accuracy improvement on RF compared to $n$ = 100, but increase 10.26 sec of computational time. In other words the improvement decreases as the number of decision trees increases. Therefore, the number of trees for the RF algorithm was set to 100 considering the benefit in prediction performance from learning additional trees in this study.

**Table 7.** Multi-classification performance with the number of tree selected.

| | Accuracy (%) | Precision (%) | Recall (%) | F1 Score (%) | ROC AUC (%) | Training Time (s) |
|---|---|---|---|---|---|---|
| $n$ = 50 | 78.63 | 69.67 | 42.05 | 40.23 | 90.05 | 12.62 |
| $n$ = 100 | 79.19 | 79.67 | 42.22 | 40.48 | 95.70 | 26.97 |
| $n$ = 150 | 79.17 | 69.61 | 42.21 | 40.47 | 95.71 | 35.73 |
| $n$ = 200 | 79.23 | 69.75 | 42.26 | 40.54 | 95.72 | 37.23 |

The average accuracy for $N_f$ = 42 was approximately 99.82% (Table 8) and 83.51% for the binary classification results on training and testing data with RF ($n$ = 100). The accuracies (in %) associated with the optimal parameter $C$ and $\gamma$-values were obtained through a cross-validation scheme for the SVC (Table 8). The binary classification accuracy

rates for the training and testing data sets were approximately 93.64% and 81.69% (C = 1000, $\gamma$ = 0.1), respectively. The optimal parameters C = 1000 and $\gamma$ = 0.1 for SVC were examined for our experiment through the use of python GridSearchCVparam = {'C_range':(0.1, 10, 100, 1000), 'Gamma_range':(0.1, 10, 100, 1000)} and the fitting error with mean_squared_error has been analyzed with a given different values of C, $\gamma$. After analyzing the experimental results, C = 1000 and $\gamma$ = 0.1 for SVC is selected in our experiments.

**Table 8.** Binary classification accuracy when multiple features were used.

|  | 42 Features (Training/Testing) | 23 Features (Training/Testing) |
|---|---|---|
| RF | 99.82% and 83.51% | 99.65%, and 83.51% |
| SVC | 93.64% and 81.69% | 93.70% and 81.60% |

As indicated in Table 8, no differences in the precision rates were noted in the first case, whereas a small subset of features accelerated the exploration of normal and intrusive patterns. Thus, a reduced feature subset for $N_f$ = 23 was selected because the threshold value of detection accuracy was considered.

Similarly, through the use of RF on training and testing data, the multiclass classification accuracies for 10 subcategories decreased to 86.04% and 54.71%, respectively, because of the imbalanced data of threat classes, as indicated in Table 9.

**Table 9.** Multiclass classification accuracy when multiple features were used.

|  | 42 Features (Training/Testing) | 23 Features (Training/Testing) |
|---|---|---|
| RF | 86.04% and 54.71% | 84.01% and 42.07% |
| SVC | 79.18% and 71.37% | 77.86% and 62.80% |

Because of the imbalanced data of threat classes, the multiclass accuracies (%) of the SVC decreased to 79.18% and 71.37% for the training and testing data, respectively.

*4.2. Case II: Over-Sampling for Misclassification Class*

Compared with that in Table 8 for UNSW-NB15 data set, the accuracy of the multiclass classifications in Table 9 was lower because of the effects of imbalanced data. We analyzed several cases of misclassification of the training data and discovered that they primarily occurred for six minority classes: Analysis (0), Backdoor (1), DoS (2), Exploits (3), Fuzzers (4), and Reconnaissance (7). Thus, SMOTE was employed to oversample six minority classes by using the command, BorderlineSMOTE(sampling_ strategy = 'minority') for multiclass classification accuracy and compare the model performance of RF with the SVC.

In this experiment, the amount of required oversampling was set to 200%, two neighbors from the five nearest neighbors were randomly selected, and one sample was generated in the direction of each of the two neighbors. The oversampling strategy for the minority class was as follows:

Resampling strategy = {class 0: 4000, class 1: 3492, class 2: 24,328, class 3: 56,000, class 4: 36,364, class 7: 20,982}.

The SMOTE resampling process increases the prediction accuracy of the classifiers, and the multiclass classification accuracy for the RF classifier incorporating the SMOTE resampling process is higher than that of the SVC classifier (Table 10). Using the SMOTE oversampling approach with RF but not with the SVC increased the multiclass accuracy by 3–4% relative to the proposed model on training and testing experiments.

**Table 10.** Multiclass classification accuracy (23 features).

|  | SMOTE(Training/Testing) | Without SMOTE(Training/Testing) |
|---|---|---|
| RF | 87.35% and 46.34% | 84.01% and 42.07% |
| SVC | 72.87% and 61.40% | 77.86% and 62.80% |

### 4.3. Case III: Binary Classification and Multiclass Classification (CSE-CIC-IDS 2018)

Similarly, an up-to-date dataset CSE-CIC-IDS 2018 [16] is conducted in the third experiment. This data set was created by the Communications Security Establishment (CSE) and the Canadian Institute for Cybersecurity (CIC) on AWS (Amazon Web Services) in 2018. CSE-CIC-IDS 2018 contained 16,233,002 records covered over the network traffics within 10 days. It included recent known attacks for intrusion detection exercise with massive network traffic and system logs. It consists of seven types of attacks including Brute-force, Heartbleed, Botnet, DoS, SQL Injection, Web attacks, and infiltration of the network from inside. The dataset includes the captures network traffic and system logs of each machine, attack classes along with 80 features extracted from the captured traffic using CICFlowMeter-V3. About 17% of the instances were malicious traffics for 7 major attacks which can be summarized in five attack categories including Brute-force, Bot, DoS, SQL Injection, and infiltration. First, 80 features of CSE-CIC-IDS 2018 were ranked using Equations (5)–(8), the set of reduced features selected from the top 36 of the 80 total features using the C4.5 approach are selected (Table 11).

**Table 11.** Top 36 features by weight. (CSE-CIC-IDS 2018).

| Feature | Weighting | Rank | Feature | Weighting | Rank |
|---|---|---|---|---|---|
| Fwd Seg Size Min | 0.0777 | 1 | Pkt Size Avg | 0.0189 | 19 |
| Init Fwd Win Byts | 0.0746 | 2 | Tot Fwd Pkts | 0.0178 | 20 |
| Fwd Pkt Len Max | 0.0503 | 3 | Bwd Seg Size Avg | 0.0170 | 21 |
| TotLen Fwd Pkts | 0.0467 | 4 | Fwd IAT Max | 0.0153 | 22 |
| Subflow Fwd Byts | 0.0453 | 5 | Fwd Pkt Len Std | 0.0149 | 23 |
| Fwd Header Len | 0.0425 | 6 | Bwd Pkt Len Mean | 0.0145 | 24 |
| Flow Pkts/s | 0.0393 | 7 | Bwd Header Len | 0.0144 | 25 |
| Fwd Pkts/s | 0.0337 | 8 | Pkt Len Max | 0.0144 | 26 |
| Init Bwd Win Byts | 0.0318 | 9 | Bwd Pkt Len Std | 0.0135 | 27 |
| Fwd Seg Size Avg | 0.0309 | 10 | Subflow Bwd Byts | 0.0131 | 28 |
| Bwd Pkts/s | 0.0299 | 11 | Flow IAT Max | 0.0130 | 29 |
| Fwd Pkt Len Mean | 0.0279 | 12 | Fwd IAT Mean | 0.0130 | 30 |
| Subflow Fwd Pkts | 0.0236 | 13 | Bwd Pkt Len Max | 0.0113 | 31 |
| Flow Duration | 0.0208 | 14 | Subflow Bwd Pkts | 0.0104 | 32 |
| Pkt Len Var | 0.0203 | 15 | Flow IAT Mea | 0.0098 | 33 |
| Fwd IAT Tot | 0.0201 | 16 | Pkt Len Mean | 0.0097 | 34 |
| Pkt Len Std | 0.0193 | 17 | TotLen Bwd Pkts | 0.0088 | 35 |
| Tot Bwd Pkts | 0.0189 | 18 | PSH Flag Cnt | 0.0084 | 36 |

First, 80% of the data files are used for training, while 20% are used for testing the model. Analyses were conducted using RF ($n$ = 150), and the SVC to evaluate the performance of the proposed model on CSE-CIC-IDS 2018. First, filtered out four descriptive metrics (unnecessary features) from 80 features in the classification process, i.e., label (output), src_port, dst_port, timestamp, protocol. According to the results, the binary classification accuracies for 99.98% and 97.05% for $N_f$ = 36 on the training and testing data, respectively (Table 12).

**Table 12.** Binary classification accuracy (CSE-CIC-IDS 2018).

|  | 76 Features (Training/Testing) | 36 Features (Training/Testing) |
|---|---|---|
| RF | 100.00%/100.00% | 99.98%/97.05% |
| SVC | 97.86%/97.90% | 96.68%/94.72% |

In an evaluation of multiclass classification for six subcategories, the average accuracies for the RF and the SVC were listed as shown in Table 13.

**Table 13.** Multiclass classification accuracy (CSE-CIC-IDS 2018).

|  | 76 Features (Training/Testing) | 36 Features (Training/Testing) |
|---|---|---|
| RF | 96.87%/88.16% | 96.53%/89.38% |
| SVC | 94.28%/84.38% | 92.46%/81.58% |

*4.4. Method Comaprison*

4.4.1. Accuracy Comparison

In this sub-section, the attributes of the proposed scheme are compared with that of the recent studies, as shown in Table 14. Practically, it is difficult to exactly compare the experimental results assocaited on different intrusion detection data sets, because the quality of original data set (affected by imbalanced ratio of data set) is different. For example, in [21,25], the studies achieved very high classification accuracy on both KDDCup 99 and NSL-KDD datasets, but not with the UNSW-NB15 dataset in [14,17]. As described in Table 2, the serious data imbalanced problems existed in the UNSW-NB15 dataset. In [21], experiments on KDD Cup 99 dataset show that the classification accuracy of random forest algorithm has reached 92.39%, which is higher than other classification methods, such as J48, LibSVM, NaiveBayes, Bagging, and AdaboostM1. Moreover, the accuracy of the RF combined with the SMOTE has increased from 92.39% to 92.57%, after over-sampling the samples for the minority classes including probing, U2R, and R2L. Overall, the experiment reports in [21] that are consistent to that in Table 10 on UNSW-NB15 dataset, i.e., the SMOTE can improve the classification effect of minority classes for imbalanced datasets on both the KDD Cup 99 and the UNSW-NB15 data sets.

From Table 14, it shows that the multi-classification accuracy of the scheme proposed in this study is higher than that of [14,17] on the UNSW-NB15 dataset. In contrast to [14,17], proposed approach has advantages on better multi-class classification accuracy on UNSW-NB15 dataset by incorporating with C4.5 DT algorithm to the model overfitting prevented from inputting additional variables to avoid learning the noise in the training data. Moreover, Table 14 indicates that the proposed approach achieved the binary accuracy close to those of [32] on CIC-IDS 2018. However, the multi-classification accuracy was 96.53% is slightly lower than that of [19].

4.4.2. Robustness of Proposed Model

To highlight the robustness of their proposed system against model underfitting and overfitting, the experimental results are summarized as following.

Imbalanced Data Handling

In the experiment, the SMOTE associated with IR metric can detect data imbalance situation by applying Equations (1) and (2). In our case, underfitting occurs when a model does not fit the input data samples enough for six minority classes which tend to decrease classification accuracy. From Table 10, the SMOTE increased over 3.34% and 4.27% of the average classification accuracy of the models on training and testing data set that are consistent to reports in [19].

**Table 14.** Performance comparison of recent studies.

| Author | Experiment Scheme/(Dataset) | Accuracy Type | Classification Accuracy |
|---|---|---|---|
| Zong, Chow, Susilo (2018) [14] | Six ML algorithms +SMOTE (UNSW-NB15) | Multi-classification (10 categories) | 85.78% |
| Tan et al. (2019) [21] | RF+SMOTE (KDDCup99) | Multi-classification (4 categories) | 92.57% |
| Karatas, Demir, Sahingoz (2020) [19] | Six ML algorithms +SMOTE (CIC-IDS 2018) | Multi-classification (6 categories) | Total: 99.34% 99.21% (original data) 99.35% (re-sampled data) for RF learner |
| Huancayo Ramos et al. (2020) [32] | Five ML algorithms (CIC-IDS 2018) | 2-class (Benign or Bot) | • 99.99% accuracy for RF and DT learners (CIC-IDS 2018) |
| Kasongo, Sun (2020) [17] | Five ML algorithms +XGBoost (UNSW- NB15) | 2-class/Multi-classification (10 categories) | • 2-class: 90.85% for DT. • Multiclass: 67.57% for DT. |
| Wu et al. (2021) [25] | GBDT+SMOTE (NSK-KDD) | Multi-classification (4 categories) | 99.72%(training set) 78.47%(testing set) |
| Proposed model | RF+SMOTE+C4.5 (UNSW-NB15) | 2-class | 99.65% |
| | | Multi-classification (10 categories) | 87.35% |
| | RF+SMOTE+C4.5 (CIC-IDS 2018) | 2-class | 99.98% |
| | | Multi-classification (6 categories) | 96.53% |

Removing Extra Features to Reduce the Risk of Overfitting

From Tables 8 and 9, we observed that users can select fewer features ($N_f$ = 23) and retain the close accuracy with $N_f$ = 42 in our experiment that made the model more flexible and reduced the risk of overfitting. In this experiment, the model overfitting is prevented from inputting additional variables to avoid learning the noise in the training data, thus causing it to decrease the computational costs and negatively impacts on the performance of the model on new data inputs.

## 5. Conclusions

This paper presents an intrusion detection model that incorporates an RF classifier with a SMOTE resampling policy to enhance the precision of the multiclass classification model by oversampling minority classes. Moreover, the proposed approach minimizes classification errors through the use of balanced data and a set of reduced features to accelerate intrusion detection. Overall, the results indicate that the precision of the proposed model for imbalanced data in intrusion detection analysis is higher than that of classifier [14,17] on UNSW-NB15 dataset.

Although SMOTE techniques incorporating RF have been proposed for intrusion detection with imbalanced data, practical challenges of using the resampling process exist. For example, the model's classification speed may be reduced if it is underfitted on imbalanced data by reducing the number of data points of majority classes. Moreover, the proposed scheme must be integrated into the RF+SMOTE module of NIDS. The scalability

challenge of large, high-speed, and complex networks for intrusion detection will be addressed in a future study.

**Author Contributions:** Conceptualization, P.W. and K.-M.C.; methodology, H.-C.L.; resources, P.W.; formal analysis, H.-C.L.; data curation, H.-C.L.; writing—original draft, Z.-Y.Y. and W.-H.L.; writing—review and editing, P.W.; software, H.-C.L.; validation, H.-C.L. and W.-H.L.; visualization, H.-C.L. and Z.-Y.Y.; project administration, P.W.; funding acquisition, P.W. All authors have read and agreed to the published version of the manuscript.

**Funding:** This research was funded by the Ministry of Science and Technology of Taiwan under Grant Nos. MOST 110-2410–H-168-003, and Taiwan's Ministry of Education (MOE) under Grant No. MOE 2000-109CC5-001.

**Institutional Review Board Statement:** Not applicable.

**Informed Consent Statement:** Informed consent was obtained from all subjects involved in the study.

**Data Availability Statement:** The data presented in this study are available on request from the corresponding author.

**Acknowledgments:** This work was jointly supported by Taiwan's Ministry of Science and Technology under Grant No. MOST 110-2410–H-168-003 and Taiwan's Ministry of Education (MOE) under Grant No. MOE 2000-109CC5-001.

**Conflicts of Interest:** The authors declare no conflict of interest.

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
