# Peer review of "Ensemble Learning for Threat Classification in Network Intrusion Detection on a Security Monitoring System for Renewable Energyâ€"

_applsci, doi:10.3390/app112311283_

Round 1
Reviewer 1 Report
Please find the attached file.

Author Response
The responses to the comments from the reviewers
Journal: Applied Sciences (ISSN 2076-3417) Ref: applsci-1472218
Title: Ensemble Learning for Threat Classification in Network Intrusion Detection on a Security Monitoring System for Renewable Energy
Reviewer 1
The author has presented an intrusion detection model that incorporates an RF classifier with a SMOTE resampling policy to enhance the precision of the multiclass classification model by oversampling minority classes. Moreover, the proposed approach minimizes classification errors through the use of balanced data and a set of reduced features to accelerate intrusion detection. Overall, the results indicate that the precision of the proposed model for imbalanced data in intrusion detection analysis is higher than that of the SVC. However, I have some suggestions as follows:
Q1. The English writing needs to improve. There are some issues related to sentence formation, phrase usage, grammar, and article.
Response 1: The English writing of manuscript revision has been sent it to a native English speaker for editing service and grammar modification in my work. Minor typing mistakes and grammatical errors have corrected in revised article. (in brown color)
Q2. The references are out-of-date. Authors should add more recent studies and compare the proposed scheme with the existing schemes. The author should consider the following recent and closely related papers to highlight their unique contribution and performance achievements.
1.Tao, Wu, Fan Honghui, Zhu Hong Jin, You CongZhe, Zhou HongYan, and Huang Xian Zhen. Intrusion Detection System Combined Enhanced Random Forest With Smote Algorithm. (2021).
- Tian, Luyao, and Yueming Lu. An Intrusion Detection Model Based on SMOTE and Convolutional Neural Network Ensemble. In Journal of Physics: Conference Series, vol. 1828, no. 1, p. 012024. IOP Publishing, 2021.
3.Tan, Xiaopeng, Shaojing Su, Zhiping Huang, Xiaojun Guo, Zhen Zuo, Xiaoyong Sun, and Longqing Li. 2019. Wireless Sensor Networks Intrusion Detection Based on SMOTE and the Random Forest Algorithm, Sensors 19, no. 1: 203. https://doi.org/10.3390/s19010203
4.Popoola, Segun I., Bamidele Adebisi, Ruth Ande, Mohammad Hammoudeh, Kelvin Anoh, and Aderemi A. Atayero 2021. SMOTE-DRNN: A Deep Learning Algorithm for Botnet Detection in the Internet-of-Things Networks, Sensors 21, no. 9: 2985. https://doi.org/10.3390/s21092985
- Jiang, Hui, Zheng He, Gang Ye, and Huyin Zhang. Network intrusion detection based on PSO-XGBoost model. IEEE Access 8 (2020): 58392-58401.
6.Yang, Jun, Yiqiang Sheng, and Jinlin Wang. A GBDT-Paralleled Quadratic Ensemble Learning for Intrusion Detection System. IEEE Access 8 (2020): 175467-175482.
Also, the authors should add some other most recent studies to compare the proposed scheme.
Response 2: The technique reviews for threat classification in network intrusion detection has updated with relevant and recent papers focused on the fields in the Table 3. Also, added mentioned 6 papers and extra recent papers as important references to References.
Q3. In the experiment, the author has used only the UNSW-NB15 dataset. However, there are some more severe attacks like botnet attack, web attack, probing, port scan, U2R, R2L, etc. Thus, the author should use some Up-to-Date Dataset to re-experiment their proposed model with the other latest dataset and severe network attacks. The author may get some idea from Table 4 and Table 5 of the following paper:
Karatas, Gozde, Onder Demir, and Ozgur Koray Sahingoz. Increasing the performance of machine learning-based IDSs on an imbalanced and up-to-date dataset. IEEE Access 8 (2020): 32150-32162.
Response 3: We extended an experiment with an up-to-date intrusion detection dataset, namely CSE-CIC-IDS2018 used in Karatas, Gozde, Onder Demir, and Ozgur Koray Sahingoz. The experiments are re-analysed and experimental results are summarized as shown in Sec 4.2 Case III.
Q4. The author has utilized Synthetic Minority Oversampling Technique (SMOT). However, there is no prior key explanation about SMOT which is necessary for the general ML audience. Include the SMOT algorithm too.
Response 4: Thanks for valuable comments. The description of SMOT algorithm is added to Sec 2 (after Fig.2)
Q5. In the current version, the preliminaries, the pre-processing steps, feature extraction, method, etc are merged up which may create confusion while reading out the paper. Thus, the author should re-write the paper by clearly separating the required preliminary section, pre-processing section, proposed model, experimental analysis etc.
Response 5: Thanks for valuable comments. The original manuscript would like to tell every detail context from the authors that causes to too confusing for most readers. For clarity, we re-write the paper as follows.
- Omit partial pictures and record the experimental results to the corresponding Tables.
- As illustrated in Figure 5, we revised the paper as follows:
The threat analysis process comprises three subprocesses: data preprocessing (including resampling process and feature selection) model training & optimization, and performance evaluation for intrusion detection of network attacks as follows.
Step 1. Data preprocessing
Step 1.1 Normalisation
Step 1.2 Resampling strategies
Step 1.3 Feature selection
Step2. Model training & optimization
Step 3. Performance evaluation
Q6. The author claimed that they have proposed modified Random Forest algorithm. But there is no such clear presentation about the modification or how the model is being modified and how stable was that?
Response 6: Thanks for valuable comments.
To point out the revision of Random Forest algorithm, the following description is added to motives of this study as follows. (Sec.1)
“Eliminate the irrelevant and unwanted features from the dataset in intrusion detection causes to possibly faster and more accurate detection. Consequently, feature selection scheme C4.5 is incorporated with RF to reduce the number of input variables to mitigate overfitting in developed predictive models. ”
Also, the following description is added to primary contributions of this study as follows.
“To improve the performance of random forests, the RF is incorporated with C4.5 algorithm to dimension reduction of training data that accelerates the training time of high-dimensional data in the model training. ”
.”
Q7. The author should include a pipeline structure or flowchart of their proposed model to provide a more clear understanding.
Response 7: Thanks for valuable comments. The flowchart of our proposed model is depicted in Figure 5. To improve readability of proposed model, we revised the context of the implementation and the analysis step by step from Step 1. Data preprocessing to Step 3. Performance evaluation followed by Figure 5. Diagram of Developed Model.
Q8. There are lots of steps, phases which are kind of mixed up. Re-arrange the steps or phases with separate and appropriate section heading and with a visual diagram if possible.
Response 8: Thanks for valuable comments. We revised the context of the implementation and the analysis step by step from Step 1. to Step 3. according to the process in Figure 5.
Q9. Instead of figure 6, 8, 9, and 10, you can use tabular form to represent the necessary parameters of your experiment. Please skip the screenshot of your experiment
Response 9: Thanks for valuable comments. We used tabular form to represent the required parameters shown in Tables 6 in our experiment. For clarity, we omitted the supplementary diagrams in the processing process and experimental results are recorded in Tables 7-12.
Q10. In the experiment, the author has set the optimal parameters C = 1000, and gamma = 0.1 only. The experiment analysis should consider the findings for other different values of C and gamma
Response 10: Thanks for valuable comments. The selection of optimal parameter is described as follows.
The optimal parameters C = 1000 and g = 0.1 for SVC were examined thru the use of python GridSearchCVparam = {'C_range':(0.1, 10, 100, 1000), 'Gamma_range':(0.1, 10, 100, 1000)} and the fitting error with mean_squared_error. After analysed the experimental results with a given different values of C, g, C = 1000, and g = 0.1 for SVC is selected in our experiments.
Q11. From Table 8, 10, 11, and 12, the overall performance is not so good comparing with some recent studies. Thus, there is a scope to improve the overall methodology and the obtained performances.
Response 11: Thanks for valuable comments.
- The low multi-classification accuracy for the UNSW-NB15 dataset presented in Table 8, 10, 11, and 12, the main reason is the low quality (data imbalanced) of training data existing in the UNSW-NB15 dataset shown in Table 2. The data imbalanced existing in UNSW-NB15 dataset in nature, it cannot increase the multi-classification accuracy of model.
- Comparing with recent studies, most existing reports used the average classification accuracy on NSL-KDD, and KDD Cup 99 datasets instead of UNSW-NB15 dataset. Basically, it cannot compare on the same basis.
- Thus, added the imbalanced data processing for UNSW-NB15 dataset in the study. The resampling process of the SMOTE algorithm is proposed to reduce the overfitting of imbalanced data by modifying the number of instances for misclassification sub-class in minority class. The experimental results show that the proposed scheme increased the multiclass accuracy by 3%–4% on training and testing experiments using the SMOTE oversampling approach with RF.
Q12. Table 12 shows the comparison with some existing models only. But, the comparison should include the state-of-art also. The author should re-write the comparison section with some more recent studies stating the state-of-art citation number.
Response 12: We re-wrote 4.3 method comparison section with some more recent studies with the performance comparisons
Q13. The author should include a section to show the robustness of their proposed system against model underfitting and overfitting to claim the practicality of their model.
Response 13: we added a subsection 4.4.2 Model robust to highlight the robustness of their proposed system against model underfitting and overfitting.
1) Imbalanced data handling
In the experiment, the SMOTE associated with IR metric can detect data imbalance situation by applying equations (1)-(2). In our case, underfitting occurs when a model does not fit the input data samples enough for six minority classes which tend to decrease classification accuracy. From Table 10, the SMOTE increased over 3.34% and 4.27% of the average accuracy of the models on training and testing data set that are consistent to reports in [21]
2) Removing extra features to reduce the risk of overfitting
From Tables 8 and 9, we observed that users can select fewer features (Nf = 23) and retain the close accuracy with Nf = 42 in our experiment that made the model more flexible reduced the risk of overfitting. In this experiment, the model overfitting is prevented from inputting additional variables to avoid learning the noise in the training data, thus causing it to decrease the computational costs and negatively impacts on the performance of the model on new data inputs.
Q14. What is the novel contribution of the proposed system with respect to the following work? Ellaborate.
Tan, Xiaopeng, Shaojing Su, Zhiping Huang, Xiaojun Guo, Zhen Zuo, Xiaoyong Sun, and Longqing Li. 2019. Wireless Sensor Networks Intrusion Detection Based on SMOTE and the Random Forest Algorithm, Sensors 19, no. 1: 203. https://doi.org/10.3390/s19010203.
Response 14: The performance comparisons between our study with Tan et al. (2019) [20] are listed as Table 14.
Table 14. Performance comparison of recent studies
Author |
Experiment scheme/(dataset) |
Accuracy type |
Classification accuracy |
|
Zong, Chow, Susilo (2018) [14] |
Six ML algorithms +SMOTE (UNSW-NB15) |
Multi-classification (10 categories) |
85.78 % |
|
Tan et al. (2019) [20] |
RF+SMOTE (KDDCup99) |
Multi-classification (4 categories) |
92.57% |
|
Karatas, Demir, Sahingoz (2020) [21] |
Six ML algorithms +SMOTE (CIC-IDS 2018) |
Multi-classification (6 categories) |
Total: 99.34% 99.21%(original data) 99.35(re-sampled data) for the RF learner |
|
Kasongo, Sun.(2020) [24] |
Five ML algorithms +XGBoost (UNSW- NB15)
|
2-class (Benign or Bot) |
· 99.99% accuracy for RF and DT learners (CIC-IDS 2018) |
|
Kasongo, Sun (2020) [25] |
Five ML algorithms +XGBoost (UNSW- NB15)
|
2-class / Multi-classification (10 categories) |
· 2-class: 90.85% for DT. · Multiclass: 67.57% for DT. |
|
Wu et al. (2021) [26] |
GBDT+SMOTE (NSK-KDD) |
Multi-classification (4 categories) |
99.72%(training set) 78.47%(testing set) |
|
Proposed model |
RF+SMOTE+C4.5 (UNSW-NB15) |
2-class |
99.65% |
|
Multi-classification (10 categories) |
87.35% |
|||
RF+SMOTE+C4.5 (CIC-IDS 2018) |
2-class |
99.98% |
||
Multi-classification (7 categories) |
96.53% |
|||
From Table 14, we obtained that the following observations regarding our study with [20]
- The same points: Random forests incorporating with SMOTE to solve the imbalanced data problems in intrusion detection.
- The different points:
- KDDCup99 dataset [20] vs. UNSW-NB15 dataset (our study).
Practically, it is difficult to exactly compare the experimental results assocaited on different intrusion detection data sets, because the quality of original data set (affected by imbalanced ratio of data set) is different. As described before, the series data imbalanced problems existed in the UNSW-NB15 dataset (Table 2).
Thus, in [20], the authors achieved very high classification accuracy (92.39%) on KDDCup 99 datasets, but not with the UNSW-NB15 dataset in [14] (85.78 %).
To improve the performance of random forests, the RF is incorporated with C4.5 algorithm in our study to reduce the input number of training data for feature selection that accelerates the training time of high-dimensional data in the model training.
3) The novel contribution of the proposed model:
From Table 14, it shows that the multi-classification accuracy of the scheme for 10 subcategories proposed in this study (87.35%) is slightly higher than 85.78 % of [14] on the UNSW-NB15 dataset.
In contrast to [14, 20], proposed approach incorporates with C4.5 algorithm to the model overfitting prevented from inputting additional variables to avoid learning the noise in the training data.

Reviewer 2 Report
I liked your article.
The material is not as simple as it might seem at first glance.
The advantages of the study are an integrated approach to data analysis and the aggregation of various methods with subsequent estimates.
Comment:
1) the list of references is drawn up with technical errors (lack of dots, spaces where they are necessary);
2) I consider the link to Wikipedia in clause 12 of the bibliography to be inappropriate (!), It should be replaced with a more authoritative one;
3) you should also carry out the technical editing of the text again; in particular - formulas are the same elements of the text as ordinary sentences, after them it is necessary to put commas and periods where necessary; for example, why after formulas (9) and (10) there are commas, not periods).
Author Response
I liked your article. The material is not as simple as it might seem at first glance.
The advantages of the study are an integrated approach to data analysis and the aggregation of various methods with subsequent estimates.
Comment:
Q1. The list of references is drawn up with technical errors (lack of dots, spaces where they are necessary);
Response 1: Thanks for valuable comments. The technical errors of references were fixed.
Q2. I consider the link to Wikipedia in clause 12 of the bibliography to be inappropriate (!), It should be replaced with a more authoritative one;
Response 2: Thanks for valuable comments. The clause 12 of the bibliography for Gradient Boosting approach is replaced by Ke, G.; Qi, Meng, Q.; Finley,T.; Wang, T.;Chen,W.;Ma, W.;Ye, Q.;Liu,T.Y. LightGBM: A Highly Efficient Gradient Boosting Decision Tree, 31st Conference on Neural Information Processing Systems (NIPS 2017), Long Beach, CA, USA, pp.1-9.
Q3. You should also carry out the technical editing of the text again; in particular - formulas are the same elements of the text as ordinary sentences, after them it is necessary to put commas and periods where necessary; for example, why after formulas (9) and (10) there are commas, not periods).
Response 3: Thanks for valuable comments.
We carry out the technical editing of the text again to rewrite the formulas (4)-(7) and fix two errors in original formulas (6) and (7).

Round 2
Reviewer 1 Report
Comments to the Authors:
The author has revised the paper. However, there are some findings:
- In Table 2, remove the comma after “Analysis”.
- At line no. 441, correct the spelling of “Through”.
- At line no. 441 and 442, apply the space after the ‘colon’ operator. The full paper consists of such mistakes in multiple places.
- The title of section 4.4.2 should be ‘Robustness of Proposed Model’.
- There are some good explanations in the cover later (near the end) that are missing from the comparative discussion between the proposed model and [20]. Include that accordingly.
Author Response
The responses to the comments from the reviewers
Journal: Applied Sciences (ISSN 2076-3417) Ref: applsci-1472218
Title: Ensemble Learning for Threat Classification in Network Intrusion Detection on a Security Monitoring System for Renewable Energy
Round 2
Q1. In Table 2, remove the comma after “Analysis”.
Response 1: Thanks for valuable comments. Corrected it.
Q2. At line no. 441, correct the spelling of “Through”.
Response 2: Corrected it.
Q3. At line no. 441 and 442, apply the space after the ‘colon’ operator. The full paper consists of such mistakes in multiple places.
Response 3: Corrected it and checked the manuscript again. Thanks.
Q4. The title of section 4.4.2 should be ‘Robustness of Proposed Model’.
Response 4: Corrected it.
Q5. There are some good explanations in the cover later (near the end) that are missing from the comparative discussion between the proposed model and [20]. Include that accordingly.
Response 5: The comparative discussion between the proposed model and [20] are summarized as follows.
In [20], experiments on KDD Cup 99 dataset show that the classification accuracy of random forest algorithm has reached 92.39%, which is higher than other classification methods, such as J48, LibSVM, NaiveBayes, Bagging, and AdaboostM1. Moreover, the accuracy of the RF combined with the SMOTE has increased from 92.39% to 92.57%, after over-sampling the samples for the minority classes including probing, U2R, and R2L. Overall, the experiment reports in [20] that are consistent to that in Table 10 on UNSW-NB15 dataset, i.e, the SMOTE can improve the classification effect of minority classes for imbalanced datasets on both the KDD Cup 99 and the UNSW-NB15 data sets.
